# Clinical Trials, Potential Mechanisms, and Adverse Effects of Arnica as an Adjunct Medication for Pain Management

**DOI:** 10.3390/medicines8100058

**Published:** 2021-10-09

**Authors:** Amanda G. Smith, Victoria N. Miles, Deltrice T. Holmes, Xin Chen, Wei Lei

**Affiliations:** 1Department of Pharmaceutical and Administrative Sciences, Presbyterian College School of Pharmacy, Clinton, SC 29325, USA; agsmith@presby.edu (A.G.S.); vnmiles@presby.edu (V.N.M.); 2Department of Biology, College of Art and Sciences, Presbyterian College, Clinton, SC 29325, USA; dtholmes@presby.edu; 3Department of Pharmaceutical Sciences, College of Pharmacy and Health Sciences, Campbell University, Buies Creek, NC 27506, USA; xchen@campbell.edu

**Keywords:** Arnica, pain, herbal medication, alternative therapy, inflammation

## Abstract

Arnica has traditionally been used in treating numerous medical conditions, including inflammation and pain. This review aims to summarize the results of studies testing Arnica products for pain management under different conditions, including post-operation, arthritis, low back pain, and other types of musculoskeletal pain. Based on data from clinical trials, Arnica extract or gel/cream containing Arnica extract shows promising effects for pain relief. These medical benefits of Arnica may be attributed to its chemical components, with demonstrated anti-inflammatory, antioxidant, anti-microbial, and other biological activities. In conclusion, Arnica could be an adjunct therapeutical approach for acute and chronic pain management.

## 1. Introduction

Pain is a serious health concern and affects more than 100 million people in the United States [1]. Pain reduces the quality of life of patients, and treatment of pain is costly [2]. There are various underlying pathologies associated with pain, such as comorbid conditions, injury, medical treatment, inflammation, neuropathy, and idiopathic pain [2]. It is widely accepted that chronic pain results from a complex interaction of physiological, psychological, and social influences [3]. The multi-faceted nature of pain complicates strategies for management [3]. Further still, the most widely utilized treatment options, such as non-steroidal anti-inflammatory drugs and opioid analgesics, show clinical efficacy in relieving pain; however, the adverse effects associated with those medications limit their use for pain management [4]. Therefore, discovering alternative therapies for pain management is in high demanded.

Arnica belongs to the Asteraceae family, a group of flowering plants containing several well-known members, such as marigold, daisy, and chamomile [5]. Arnica has been used in traditional and homeopathic medicine for centuries in Europe and North America [6]. Multiple species within this genus have been documented and studied for medical potential. *A. montana* and *A. chamissons* are native to Europe, while *A. acaulis*, *A. cordifolia*, *A. fulgens*, and *A. sororia* are native to North America [7]. Many species are used in homeopathic and traditional medicine practices, although *A. montana* is the most commonly used species in commercially produced products [8].

Arnica has been used in homeopathic medicine for centuries for dozens of pathological conditions, including joint and muscle pain, inflammation, and arthritis [8]. Arnica has various pharmacologic activities, including anti-inflammatory, analgesic, anti-microbial, antirheumatic, antiarthritic, and antineoplastic activities [9]. Although Arnica is still used for these alternate indications, ingestible formulations are not utilized in modern therapies due to their cytotoxic properties [8]. Some homeopathic treatments may use highly diluted Arnica preparations for internal uses, although topical administration is most common [10].

Treatment is available in the form of several homeopathic commercial products, including gels, creams, roll-ons, teas, pellets, liquids, and tablets [11]. These products are widely available in retail pharmacies ranging in strengths from 1 X to 30 X. Homeopathic medicine is dosed based on ten-fold dilutions, so a 30 X product has been diluted in a 1:10 ratio thirty times [12]. Topical preparations are more widely available, but there are ingestible products on the market. Marketing for these products is targeted at various patients, including those with chronic back pain, surgical wounds, bruising, cough, and arthritis [13].

The German Commission E approved the use of Arnica flower products to treat hematomas, dislocations, contusions, edema, rheumatic joint and muscle pain, inflammation of the mouth and throat, insect bites, and superficial phlebitis [8]. The European Union has authorized the use of several Arnica-containing preparations—notably tinctures and liquid flower extracts—for external use only, and Arnica products are most often composed of *A. montana* due to the cultivation of this species [8]. In the United States, Arnica is available in natural products, although varying concentrations can occur in these formulations due to less strenuous labeling and testing requirements [14]. In addition to pharmaceutic preparations, it is also used in cosmetic products, hair tonics, anti-dandruff products, and as natural flavorings in foods and beverages [11].

This paper aims to collect evidence for using Arnica as an alternative approach for pain management. We will summarize the findings from clinical trials testing Arnica on pain relief. We will also discuss the potential mechanisms of action, safety, and adverse effects of Arnica for pain management.

## 2. Arnica for Pain Management

As mentioned above, Arnica has a long history of being used for pain relief. In the last few decades, clinical studies have been conducted to investigate the activity of Arnica on pain under different conditions, such as post-operation and with arthritis. The commonly used forms of Arnica are a diluted hydroalcoholic extract of flower heads or fresh plants, creams, ointments, and gels containing Arnica extract or Arnica oil [11].

### 2.1. Post-Operative PAIN

Post-operative pain is one of the most common types of acute pain to occur after an operation, and effective management of pain is essential following a surgical procedure [15].

Arnica has mainly been studied on patients suffering from post-operative pain. Pain after different surgical processes affects patients’ quality of life and recovery, along with increasing the cost [16]. Arnica has shown mixed effects on post-operative pain. Treatment with either Arnica tablets or herbal Arnica ointment for two weeks significantly reduced pain after hand surgery compared to placebos [17]. In another study, after varicose vein surgery, patients who were treated with Arnica showed a trend toward reducing pain [18]. Brinkhaus et al. investigated the effectiveness of Arnica on post-operative pain after arthroscopy, artificial knee joint implantation, or cruciate ligament reconstruction [19]. They found that Arnica only showed significant beneficial improvement in patients after cruciate ligament reconstruction. Dental surgical procedures cause pain and other postsurgical sequels [20], and trials have been conducted to investigate the activity of Arnica on those symptoms. Mawardi et al. reported that the systemic use of Arnica significantly reduced pain scores after dental extraction compared to the placebo group [21]. Similar results were found in another trial conducted on patients following tonsillectomy [22]. Patients treated with two tablets six times during the first post-operative day, and then two tablets twice a day for the next seven days, displayed a significant decrease in pain scores after their tonsillectomy [22]. However, Kaziro and colleagues reported the opposite effect after using Arnica as a treatment for patients undergoing the removal of impacted wisdom teeth [23]. They found that patients who received the Arnica treatment suffered greater pain compared to patients treated with a placebo [23]. Several other trials reported that therapy with Arnica had no evidence of effects on pain relief after surgical procedures, even though it showed improvement on other symptoms related to the operation [16,24,25,26]. Patients undergoing elective surgery for carpal tunnel syndrome were treated with Arnica or a placebo for seven days before surgery and fourteen days after surgery [24]. Compared to the placebo group, the results showed no improvement for Arnica in reducing patients’ post-operative pain [24,27]. Cornu et al. also found that Arnica combined with Bryonia did not relieve pain after the cardiac surgery [25]. Barlow et al. reported that Arnica showed no evidence of an effect on pain after knee surgery [16]. In addition, Arnica ointment 10% did not improve any outcomes, including pain, after undergoing upper blepharoplasty [26].

In another clinical trial, Karow et al. compared the effect of Arnica to that of diclofenac on patients who received hallux valgus surgery. They found that Arnica showed a modest impact on pain relief but was less toxic compared to diclofenac [28]. Procedural neonatal pain is a type of pain associated with the examination, investigation, and treatment of neonates and infants, and uncontrolled neonatal pain could affect long-term development [29,30]. There is one trial that investigated the effect of Arnica on neonatal pain. Treatment with Arnica exhibited the most efficacy on pain relief at 30 s and 5 min after the procedure compared to the control and Sol.Glucosae 25% (an analgesic for neonatal pain) [31]. In addition, treatment with Arnica reduced seroma formation and opioid intake in patients who underwent mastectomy and reconstruction [32].

In summary, Arnica showed different effects in different trials. However, it was better tolerated compared to the commonly used medications for post-operative pain, such as diclofenac [28]. It also could reduce opioid intake for pain management after surgery [32]. However, more studies are needed to confirm the impact of Arnica on post-operative pain and its treatment.

### 2.2. Pain in Patients with Arthritis

Chronic inflammatory and noninflammatory joint diseases, such as osteoarthritis, are major causes of pain in adults [33]. A few clinical trials have tested the effect of Arnica on pain in patients with arthritis. Patients with hand osteoarthritis that was treated with Arnica extract gel for three weeks exhibited a moderate reduction in their pain score [34]. The Arnica fresh plant gel showed similar activity to the gel extract in patients with knee osteoarthritis. Pain scores were significantly reduced through treatment with the Arnica fresh plant gel for three and six weeks [35]. Another trial conducted with patients with calcific periarthritis of the shoulder found that an ointment mixture of Arnica, Acidum, and Hekla resulted in a more potent reduction in pain compared to Arnica alone, which may indicate a synergistic effect when Arnica is used with other herbs [36]. Knuesel et al. compared the effect of ibuprofen and Arnica on patients with hand osteoarthritis and found that Arnica showed comparable pain management activity to that of ibuprofen [37]. In addition, Tsintzas et al. reported that Arnica showed promising improvement in joint pain for two patients [38].

### 2.3. Low Back Pain

Low back pain is a growing health issue and has been the most common cause of work-related disability [39]. Approaches for the management of low back pain are sorely needed. Treatment with a homeopathic complex, including Arnica, Bryonia, Causticum, Kalmia, Rhus, and Calcarea, was found to significantly improve pain symptoms in patients with chronic low back pain, even though it did not change the need for conventional pain medication [40]. da Silva et al. tested a fluid extract of *Solidago chilensis*, a Brazilian Arnica, in patients with lumbago and found that two daily skin applications of a gel containing a 5% of such extract for 15 days significantly reduced the perception of pain [41]. However, the results from the trial that tested this Brazilian Arnica in low back pain patients need to be further investigated [42].

### 2.4. Other Musculoskeletal Pain

A trial tested the effect of homeopathic therapy containing Arnica 200 CH, with and without a combination of two creams containing Arnica, on pain resulting from sports post-trauma ankle sprains in amateur sport practitioners [43]. Treatment with Arnica alone or in combination with Arnica creams, for 6 and 12 days, reduced pain more efficiently compared to the control group [43]. The application of a gel cream containing 5% *Solidago chilensis* (a Brazilian Arnica extract) to the skin for 21 days reduced arm pain in patients suffering from tendinitis of the flexor and extensor tendons [44].

The effect of Arnica on exercise-induced muscle pain has been tested in healthy individuals. Topical Arnica, applied to the skin every four waking hours, reduced pain three days after a downhill run [45]. Unfortunately, leg pain 24 h after eccentric calf raises was increased through treatment with a cream containing Arnica [46].

## 3. Mechanisms of Action for Pain Management

The beneficial impact of Arnica on pain relief may be the result of its anti-inflammatory, anti-microbial, antioxidant, and immunomodulatory activities [10,47,48]. The anti-inflammatory properties of Arnica have been widely investigated in cell and animal models. The *Lychnophora passerina* (a Brazilian Arnica) crude ethanolic extract, and its ethyl acetate and methanolic fractions, reversed the lipopolysaccharides/interferon gamma-induced nitric oxide (NO) and tumor necrosis factor-alpha (TNF-α) production in J774.A1 macrophages [49]. The crude extract and fractions also increased the production of the anti-inflammatory cytokine, interleukin (IL)-10 [49]. Further, an ointment containing either 10% of the crude ethanolic extract or 20% of the fractions showed a similar effect to that of diclofenac in the carrageenan-induced paw edema mice model [49]. The anti-inflammatory activities of Arnica were also determined in human umbilical vein endothelial cells stimulated with TNF-α [50]. Treatment with Arnica diminished the expression of the intracellular cell adhesion molecule (ICAM-1) induced by TNF-α in the endothelial cells [50] and increased the anti-inflammatory macrophage population compared to pro-inflammatory macrophages in mice with injured skeletal muscles. Alfredo et al. demonstrated that a massage with Arnica gel for three days reduced the density of polymorphonuclear cells in rats with Tibialis Anterior muscle lesions, thus indicating an inflammatory response [51]. Using a rat paw edema model induced by carrageenin, Kawakami et al. found that Arnica attenuated edema and the degranulation of mast cells, and increased the lymphatic vessel diameter [52]. The methanol extract of the Arnica flower exhibited inhibitory activity on the expression of TNF-α, IL-1β, IL-6, and IL-12 in rats with collagen-induced arthritis [53].

There has been a wide variety of claims regarding the anti-microbial properties of Arnica components [54]. In traditional medicine, Arnica was routinely used for infection, especially dental and oral infections [54]. The mechanism is not understood, and there is little data on its clinical use in humans for anti-microbial activity; however, in animal models, anti-microbial properties are observed [55].

A study of cutaneous leishmaniasis in Golden hamsters found that an Arnica tincture caused slightly better resolution and wound-healing than meglumine antimonate, a standard treatment drug for leishmaniasis [55]. The synergistic anti-inflammatory properties of Arnica may also contribute to the improvement of lesion healing [6]. Another study found improvements in pododermatitis infection and edema in captive penguins treated with oral Arnica and Calcarea, without notable adverse effects [56].

The ethanol/water extract of Arnica flowers, rich in flavonoids and phenolic acids, demonstrated high, Trolox-equivalent antioxidant capacity, oxygen radical absorbance capacity, and free radical-scavenging activity [57]. This extract also prevented hydrogen peroxide-induced oxidative damage and morphological changes in fibroblast cells [57]. The radical scavenging activity was also determined in crude methanol extracts of Arnica [48].

Over 150 bioactive components have been identified in the stem and leaves, flowers, seeds, and roots of Arnica [6]. These include sesquiterpene lactones, flavonoids, volatile oils, coumarins, helenalin, carotenoids, diterpene alcohols, arnidiol, pyrrolizidine alkaloids, phenolic acids, essential oils, oligosaccharides, and lignans [6,58]. The anti-inflammatory, anti-microbial, antioxidant, and immunomodulatory activities of the chemical compounds in Arnica have been investigated in different models [59,60,61,62,63].

One source indicates that the polysaccharides, arabino-3,6-galactan-protein, and fucogalactoxyloglucan, in the Arnica flowers significantly promote phagocytosis and the production of TNF-α in immune cells [59]. Helenalin, one of the primary active components of Arnica, also expresses anti-inflammatory and immunosuppressive effects [60]. Helenalin can induce apoptosis and inhibit CD-4^+^ T cell proliferation through stabilizing p53, increasing reactive oxygen species (ROS), and suppressing nuclear translocation of the nuclear factor of activated T cells, cytoplasmic 2 (NFATc2), in activated CD-4^+^ T cells [60]. Helenalin also inhibits the activation of nuclear factor-kappa B (NF-κB) in T cells, B cells, and epithelial cells, via diminishing the phosphorylation of IκB, which promotes the dissociation of NF-κB/IκB, leading to the activation of NF-κB [61,62]. A high concentration of sesquiterpene lactones (SL)—secondary plant metabolites from Arnica suppressed IL-12 production and NF-κB activation in dendritic cells [63]. In contrast, a low concentration of SL exhibited immunostimulatory effects [63].

## 4. Safety and Toxicity

Arnica is generally used via oral administration and in a topical form [10]. The medicinal formulations used in homeopathic and commercially produced products contain diluted concentrations tolerated in oral and topical products [34]. However, dosing is not standardized across different products, resulting in variation in concentration levels between different manufacturers; this is a concern, as safety risks may be associated with higher doses [34].

Preclinical studies demonstrated the oral LD50 of the extract in mice and rats [64,65]. Paula-Freire et al. tested a range of doses of a hydroalcoholic extract (30–100 mg/kg) and found that the LD50 was 512.5 mg/kg body weight in mice [64]. Another study demonstrated that the LD50 for Arnica was 54.7 mg/kg body weight in rats [65].

In clinical trials, adverse dermatologic effects were commonly reported with topical therapies. Contact dermatitis can develop, resulting in a rash, itching, and dry skin [66]. Allergic reactions are common with topical products and are cross-reactive with plants in the Compositae and Asteraceae families [67].

As indicated previously, oral formulations pose a greater risk of toxicity and cause more adverse effects than topical formulations [10]. Oral products are more sensitive to inconsistencies in product concentrations, particularly in higher concentrations. In clinical trials, patients reported gastrointestinal effects, dry mouth, headache, drowsiness, and lethargy [24]. High concentrations are associated with gastroenteritis, vomiting, diarrhea, shortness of breath, and tachycardia [28]. Dyspnea and cardiac arrest may result [25]. Arnica also has anticoagulant and antiplatelet effects due to its coumarin content. Many clinical trials contraindicated the use of Arnica while on anticoagulant or antiplatelet drug therapies due to an increased risk of developing a bleed [11]. Arnica therapy may also lower the efficacy of antihypertensive drug therapies [11].

Due to a lack of data, combined with its toxic components, Arnica is typically considered unsafe to use during pregnancy [68]. Data from some studies suggest that topical products may be safe for use during breastfeeding [68]. One study documented hemolytic anemia and increased bilirubin in a nine-year-old breastfed infant that was likely caused by maternal ingestion of Arnica [69].

## 5. Conclusions

In conclusion, Arnica presents comparable activity to standard medications, such as ibuprofen and diclofenac, for pain management under different medical conditions. It also shows fewer adverse effects along with lower costs. Arnica could be an adjunct approach for acute and chronic pain management.

## Data Availability

All the data are available within the article.

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
