# Peer review of "Clinical Trials, Potential Mechanisms, and Adverse Effects of Arnica as an Adjunct Medication for Pain Management"

_medicines, 2021, doi:10.3390/medicines8100058_

Round 1
Reviewer 1 Report
I have read this article with a great interest. It is well written. However, presentation of data is not fully satisfactory as the toxicologic data are missing. Also. The drug doses were not standardized and hence are the results incomparable between the trials. In one place reference [28], the sentence is wrongly constructed suggesting that Arnica was less effective against the pain in comparison to diclofenac. After reading the original article it appeared that Arnica was slightly better but less toxic. This should be corrected. I object against the sentence in the introduction “Further still, the most widely utilized treatment options, such as non-steroidal anti-inflammatory drugs and opioid analgesics, lack clinical efficacy in relieving pain, and the adverse effects associated with those medications limit their use 30 for pain management [4].” I think the efficacy of NSAIDs and opioids is confirmed, but these drugs are not ideal.
It is unclear what the authors mean by German Commission E.
Brinkhaus and his colleagues…. Should be Brinkhaus et al.
In the summary of the subsection on postoperative pain a new and not mentioned earlier reference [32] is presented. It should be discussed earlier.
First the Brazilian Arnica is mentioned [vers 141] and later it is redefined as “Lychnophora passerina (a Brazilian Arnica)”. Mazy I suggest to put all Latin names of the plants in italic?
I loved the subsection on the mechanism of action of Arnica.
All together, I found this paper interesting although the evidence for efficacy of Arnica is still weak and the authors did not change this with their article.
Author Response
- I have read this article with a great interest. It is well written. However, presentation of data is not fully satisfactory as the toxicologic data are missing. Also. The drug doses were not standardized and hence are the results incomparable between the trials. In one place reference [28], the sentence is wrongly constructed suggesting that Arnica was less effective against the pain in comparison to diclofenac. After reading the original article it appeared that Arnica was slightly better but less toxic. This should be corrected. I object against the sentence in the introduction “Further still, the most widely utilized treatment options, such as non-steroidal anti-inflammatory drugs and opioid analgesics, lack clinical efficacy in relieving pain, and the adverse effects associated with those medications limit their use 30 for pain management [4].” I think the efficacy of NSAIDs and opioids is confirmed, but these drugs are not ideal.
Response:
We thank the Reviewer for the constructive comments and suggestions. It is very helpful for improving the quality of this manuscript.
We have added a short paragraph to describe the LD50 values (Line 232-235).
We agree with the reviewer that the drug doses were not standardized. However, the clinical trials included in this manuscript used different Arnica products, such as extracts and cream, which contained different components. It would be greatly appreciated if the reviewer could provide some specific suggestions on this issue, and we are willing to revise the manuscript to resolve this issue.
We also revised the statements related to reference #28 and common pain medications.
- It is unclear what the authors mean by German Commission E.
Response: The German Commission E is a scientific advisory board of the Federal Institute for Drugs and Medical Devices. It provides approval of substances and products previously used in traditional, folk, and herbal medicine.
- Brinkhaus and his colleagues…. Should be Brinkhaus et al.
Response: Corrected.
- In the summary of the subsection on postoperative pain a new and not mentioned earlier reference [32] is presented. It should be discussed earlier.
Response: We added a statement to describe this study (Line 118-119).
- First the Brazilian Arnica is mentioned [vers 141] and later it is redefined as “Lychnophora passerina (a Brazilian Arnica)”. Mazy I suggest to put all Latin names of the plants in italic?
Response: Corrected.
I loved the subsection on the mechanism of action of Arnica.
All together, I found this paper interesting although the evidence for efficacy of Arnica is still weak and the authors did not change this with their article.
Reviewer 2 Report
Thank you for permitting me to review this manuscript
In this paper the authors reviewed the effect of Arnica as an alternative to other pain medication
Major concern
I think the title is misleading? nowadays treatment of pain is multimodal and I believe the title should be changed to "adjunct treatment" instead of alternative , especially the authors themselves cited multiple studies denying the role of Arnica in pain treatment. In one study Arnica was reported to be able to replace NSAID (reference 10)
For acute postoperative pain limitation could be the effect on platelet and coagulation pathway , which could be recited in this chapter .
A brief conclusion for postoperative pain is necessary as it appears to me the percentage of negative study are greater than positive studies.
If the authors have personal experience with this plant they may publish their experience as (unpublished data)
Line 159-161: please provide multiple references (PPR)
Line 181-3: PPR
Line 188-189: PPR
Line 203: PPR
Line 204-215 This paragraph need some development as it is too short with too much non explained abbreviations
Line 228: PPR
Conclusion : Again in my point of view it is is an alternative approach for NSAID only not for all pain medications
Author Response
Thank you for permitting me to review this manuscript
In this paper the authors reviewed the effect of Arnica as an alternative to other pain medication
Response: We are very grateful to this Reviewer for his/her favorable comments and suggestions on our manuscript.
Major concern
- I think the title is misleading? nowadays treatment of pain is multimodal and I believe the title should be changed to "adjunct treatment" instead of alternative , especially the authors themselves cited multiple studies denying the role of Arnica in pain treatment. In one study Arnica was reported to be able to replace NSAID (reference 10)
- Conclusion : Again in my point of view it is is an alternative approach for NSAID only not for all pain medications
Response: We revised this term in the title, abstract, and conclusion.
- For acute postoperative pain limitation could be the effect on platelet and coagulation pathway , which could be recited in this chapter .
Response: We thank the reviewer for the suggestion. Similar information was included in the “Safety and Toxicity” section.
- A brief conclusion for postoperative pain is necessary as it appears to me the percentage of negative study are greater than positive studies.
Response: A statement was added (Line 123-124).
- If the authors have personal experience with this plant they may publish their experience as (unpublished data)
Response: Thank the reviewer for the generous suggestion. Our own project is in the process, and hope that we would be able to publish the data in a separate paper.
- Line 159-161: please provide multiple references (PPR), Line 181-3: PPR, Line 188-189: PPR, Line 203: PPR, Line 228: PPR
Response: References were added.
- Line 204-215 This paragraph need some development as it is too short with too much non explained abbreviations
Response: We revised this paragraph with more details.